# The Urgent Need for Nutritional Medical Care in Geriatric Patients—Malnutrition in Nursing Homes

**DOI:** 10.3390/nu15204367

**Published:** 2023-10-14

**Authors:** Harald K. Widhalm, Maximilian Keintzel, Gerald Ohrenberger, Kurt Widhalm

**Affiliations:** 1Clinical Division of Traumatology, Department of Orthopedics and Trauma Surgery, Medical University of Vienna, 1090 Vienna, Austria; 2Clinical Division of Orthopedics, Department of Orthopedics and Trauma Surgery, Medical University of Vienna, 1090 Vienna, Austria; maximilian.keintzel@meduniwien.ac.at; 3Austrian Academic Institute for Clinical Nutrition, 1090 Vienna, Austria; office@oeaie.org; 4Haus der Barmherzigkeit Wien, Seeböckgasse 30A, 1160 Vienna, Austria; gerald.ohrenberger@hausderbarmherzigkeit.at

**Keywords:** malnutrition, hypoalbuminemia, elderly, geriatric patients, nursing home

## Abstract

Patients aged 65 and over who are accommodated in hospitals and nursing homes are at high risk of malnutrition and often show signs of it. The future relevance of this problem becomes clear, especially in view of the demographic development of the coming years and decades. In this study, the correlation between malnutrition, hypoalbuminemia, anemia, elevated CRP, and low transferrin levels, as well as mortality in seniors between 65 and 100 years, should be revealed. Therefore, the prevalence of disease-specific malnutrition (DRM), according to the criteria of the guidelines of the German Society of Nutritional Medicine (DGEM), and the prevalence of hypoalbuminemia were presented based on the data of 120 residents who were inpatients in a large Viennese nursing home between 01/2017 and 08/2020. Moreover, 86 of the inpatient residents were women and 34 were men, with a mean age of 84 years (SD: 8.7). In this examination, more than one-third of nursing home residents were malnourished. More than half of the residents were found to have low serum albumin or low transferrin saturation. However, no correlation between elevated CRP, low transferrin, or low serum albumin values and malnutrition could be established. Residents with low serum albumin or low transferrin levels, however, had a higher mortality rate. This study supports the urgent relevance of closer and individually personalized medical nutritional interventions, especially concerning hypoalbuminemic seniors aged 65 years and older.

## 1. Introduction

According to Statistics Austria, in 2005, there were 1,307,945 people in Austria over the age of 65 years, representing a percentage of around 15%. By 2020, this number had risen to 1,707,643 and already accounted for 19.1% of the total population. By 2030, according to demographic forecasts, 2,140,717 and by 2040 already 2,490,916 persons over 65 years of age—that is 23.3% and 26.4% of the population, respectively—will be living in Austria [1]. In addition, the number of inpatients in nursing homes in Austria aged 65 and over increased by a full 25% between 2015 and 2020, from 75,632 to 95,263 [1]. 

Primarily elderly people in institutions are among the high-risk groups for the development of malnutrition due to various age-related changes. Aging-related conditions such as dementia [2], sensory loss, loss of appetite [3], inflammation [4,5,6], comorbid illnesses, dysphagia, and polypharmacy increase the likelihood of weight loss or poor nutrition [7,8,9,10]. Recently published studies have confirmed that manifest malnutrition, regardless of definition, leads to increased mortality (2% in hospitals and 6% in nursing homes) [11] and, with a median of 6 days, a significantly prolonged length of stay (LOS) as well as an increase in readmission rate. Thus, it causes more costs than compared with well-nourished patients with a similar diagnosis [11,12,13,14]. For example, according to a study published in 2003, malnourished patients stayed in the hospital for 16.7 days (SD: 24.5), whereas normally nourished patients had a length of stay of 10.1 days (SD: 11.7). Hospital costs for malnourished patients were increased by up to 308.9% [15]. Schmidt et al. estimate that in Germany, costs of about 9 billion euros per year are caused by nursing measures or hospitalizations due to malnutrition [16]. S. Moramarco et al., in a study published in 2020, reported a prevalence of 67.4% of hypoalbuminemia in hospitalized patients aged 65 years or older, highlighting the role of albumin as a useful marker of malnutrition and frailty [17]. 

In 2019, X. Hong et al. also confirmed that elderly hospitalized patients who were well-nourished and had higher serum levels of total protein and albumin were less likely to become frail [18]. Furthermore, in a large-scale frailty study by Song et al., it was shown that hypoalbuminemia is a prognostic mortality factor in the elderly, whether they live in the community, are hospitalized or institutionalized. Accordingly, low albumin levels are associated with poorer recovery after acute pathologies and additionally increase the risk of death by 57% in individuals aged 65–102 years [19]. In view of the demographic development of the elderly population in the coming years and decades, the high relevance of the problem becomes apparent.

Especially in the central European region, the role of malnutrition in nursing homes has only been studied to a limited extent and is mostly significantly underestimated [1,20].

The aim of this study was to demonstrate the prevalence of malnutrition in the patient population of an Austrian nursing home and shed light on its clinical relevance. The hypothesis was that undernourished patients have an increased susceptibility to the occurrence of complications as well as an increased mortality rate.

## 2. Materials and Methods

The chosen methodological approach was a non-experimental, descriptive, quantitative cross-sectional design. The data for this study arises from the database of a large nursing home in Vienna, Austria, which has not been evaluated or analyzed so far. 

Data from patients admitted to the nursing home between 01/2017 and 08/2020 were used for this analysis. 

Exclusion criteria in the study were an inpatient stay of less than 12 months, no admission blood samples, underlying malignancies, infectious disease, and surgery performed during the inpatient stay, as well as patients younger than 65 years of age.

The patient data were submitted in already anonymized form and included only data of “older/elderly” patients with at least 65 years of age. Using the case numbers already anonymized on-site, the patient data was sorted chronologically. From the extensive database of 545 patients, 120 patients were eligible for this study. Age, sex, height, and weight at the time of admission were entered into the resulting register. In addition, if available, weight was also entered 3, 6, and 12 months after the time of admission to calculate the BMI and to determine any changes in weight and malnutrition prevalence. Laboratory parameters of the admission blood were used to further determine the health and nutritional status of each nursing home resident. This analysis included the testing of C-reactive protein (CRP), transferrin, serum albumin, and hemoglobin. Finally, the status of the nursing home residents was noted. The status of the individuals indicated whether they were deceased, discharged, or still in residential care at the time of data collection. The malnutrition status was calculated using the guidelines of the German Society of Nutritional Medicine (DGEM). The prevalence was noted after 3, 6, and 12 months of stationary stay. 

### Statistical Analysis

The open-source statistical program package R (version 4.0.4 x86) and the statistical program IBM SPSS Statistics 27 under Windows 10 were used for the analysis.

The alpha error level was set at 5% (two-sided) for each test; no adjustment was made for the alpha error. The results of the statistical tests are therefore to be understood as purely descriptive. Missing values were not replaced. The open-source statistical software package R (version 4.0.4 x86) under Windows 10 was used for the evaluation.

For unrelated categorical variables such as sex, Fisher’s exact test (Fisher) (2 × 2 tables) was used.

For unrelated ordinal variables in the case of two independent groups, such as age, BMI, and laboratory parameters, the Mann–Whitney–exact U (MWU) test was used.

Additionally, for the comparison of two independent groups, the two-sample *t*-test for independent samples was used. In the case of variance heterogeneity (checked by Levene’s test), Welch’s *t*-test was applied. In the case of a non-normal distribution of the data (test by Kolmogorov–Smirnov test with Lilliefors correction, alpha = 10%), the exact Mann–Whitney–U test was carried out (Table 1).

## 3. Measurements

### 3.1. Malnutrition

In the guidelines of the German Society of Nutritional Medicine (DGEM), “DGEM Terminology in Clinical Nutrition”, various definitions of “nutritional status with relevance for nutritional support” were reprocessed and summarized [20]. Disease-related malnutrition (DRM) was defined by the following three independent criteria: Body mass index (BMI) < 18.5 kg/m^2^ ORUnintentional weight loss > 10% in the past 3–6 months ORBMI < 20 kg/m^2^ AND unintentional weight loss > 5% in the past 3–6 months.

### 3.2. Hypoalbuminemia

Hypoalbuminemia is defined by a serum albumin level below 35 g/L. Low albumin levels are most commonly observed in elderly patients, especially those who are institutionalized and/or hospitalized [21]. Various studies have shown that a progressive decrease in serum albumin concentration between 0.08 and 0.17 g/L per year is associated with aging [21,22,23,24]. According to most studies, serum albumin is the most commonly used, cost-effective nutritional marker for risk stratification of patients during their stay in a hospital or nursing home [17,25]. Some studies also suggest that hypoalbuminemia in nursing home residents is significantly associated with malnutrition [26,27]. In this work, serum albumin values between 35 and 53 g/L are assumed to be normal values.

### 3.3. Age 

The World Health Organization (WHO) and the European Society for Clinical Nutrition and Metabolism (ESPEN) classify elderly people according to their chronological age. Persons older than 65 years of age are referred to as “elderly”, while persons older than 85 years of age are referred to as “very old” or “very elderly” [28,29]. In this paper, the above categorization of patients into “elderly” and “very elderly” is also used for statistical data analysis. 

### 3.4. CRP

Chronic inflammation and the associated elevated CRP in old age are major risk factors associated with age-related diseases such as cardiovascular disease, hypertension, diabetes mellitus, and kidney disease. In addition, a clear association with frailty could be shown. Pourhassan et al. were also able to show that there is a clear association between dietary intake and inflammation.

The levels between 0.0 and 0.5 mg/dL indicated no inflammation (=normal values) whereas levels between 0.5 and 3.0 mg/dL and >3.0 mg/dL were considered as mild inflammation and moderate to severe inflammation, respectively [30].

### 3.5. Transferrin and Hemoglobin 

The occurrence of anemia in the elderly is multifactorial, but malnourished nursing home patients appear to have a significantly higher likelihood of developing anemia [31,32].

Rondoni et al. suggested that anemia and low serum transferrin can also be surrogate markers of protein deficit and should be considered when evaluating the general nutritional status of the institutionalized elderly patient [33].

According to Emmanuel et al., 30–40% of anemia is nutrition-related [34].

In this work, transferrin levels below 200 mg/dL are considered decreased, while Hb levels below 13 mg/dL in men and below 12 mg/dL in women were considered pathological.

## 4. Results

Here, 86 (72%) of the 120 nursing home inhabitants were women, and 34 (28%) were men. The mean age of the participants included in the study was 84 SD: 8.7 years, with 56% of the patients older than 85 years. The minimum age of the participants was 65 years, and the maximum age was 100. The admission weight could be determined for 108 of the 120 patients. Body height was measured in 95 patients of the total collective. Consequently, BMI could be determined in 95 of the 120 patients.

### 4.1. BMI

The mean body mass index (BMI) of the sample was 22.8 SD: 3.56 kg/m^2^ in males and 22.8 SD: 4.365 kg/m^2^ in female participants, with the minimum BMI in females being 12.9 and the maximum 35.9 kg/m^2^. In males, the minimum BMI value was 16.7 kg/m^2^, and the maximum value was 32.1 kg/m^2^. During the further inpatient stay (after 6 and 12 months), the mean BMI did not change significantly in either male or female participants. Approximately, 23% (n = 22) had a BMI of less than 20 kg/m^2^ at admission, with 17 participants (18%) even having a BMI of less than 18.5 kg/m^2^. Here, 9 of the 90 patients had a BMI of less than 17.5 kg/m^2^ at the time of admission. 

### 4.2. Status of Patients

Between 01/2017 and 08/2020, 55 of the 120 patients (45.8%) with varying lengths of stay were deceased. Additionally, 4 patients were discharged within that same period and 61 patients (50.8%) were still in inpatient care at the time of data collection (see Table 2 and Table 3.)

### 4.3. Malnutrition

Malnutrition status could only be determined in 95 of the 120 patients, as anthropometric data were not available for the remaining 25 study participants for various reasons. According to DGEM criteria, malnutrition was present in 37.9% of nursing home residents during the first 3 months of admission (Table 4). After 6 months, 41.1% of nursing home residents could be classified as DRM according to the DGEM guidelines (Table 5). After 12 months of inpatient stay, 3 inhabitants have been deceased, with 3 other residents no longer considered malnourished after this period, according to the criteria used (Table 6). However, no significant correlation between DRM (Disease-related malnutrition) and increased mortality was found in this study (*p* = 0.200). 

### 4.4. Hypoalbuminemia (Reference: 35–53 g/L)

In 119 of 120 individuals, serum albumin could be obtained. The mean serum albumin level of the participant group at the time of admission was 33.5 g/L. Thus, low serum albumin values were found in 58.8%, i.e., 70 of the 119 patients. A mean serum albumin level of 35.0 g/L was detected in the 61 stationary patients. Thus, 51% were above the lower reference level of 35 g/L. In contrast, it has been found that deceased inhabitants only had a mean serum albumin value of 31.7 g/L at the time of admission. Hence, 40 of the 55 (71%) deceased patients had hypoalbuminemia (see Table 7. At a significance level of *p* < 0.001, the correlation between low serum albumin levels and higher mortality rates among the subjects in this study could be confirmed. 

Residents with “DRM” had a mean serum albumin level of 34.2 g/L, whereas non-malnourished subjects had a mean value of 34.4 g/L. Thus, no significant difference concerning DRM and hypoalbuminemia could be found (Table 8).

### 4.5. Transferrin (Reference: 200–360 mg/dL)

The mean transferrin level of the total collective was 197.4 SD: 46.93 mg/dL, with a minimum of 75 mg/dL and a maximum level of 360 mg/dL. 56% of nursing home residents had low transferrin values. A correlation between mortality among institutionalized persons and low transferrin levels could be confirmed (*p* < 0.001).

Deceased residents (n = 61) had a mean transferrin value of 187.3 SD: 50.9 mg/dL. The minimum and maximum values were 75 and 281 mg/dL, respectively. The inpatient group (n = 55) had a mean value of 204.57 SD: 42.6 mg/dL and values between 125 and 360 mg/dL (see Table 9). There were no significant differences between malnourished and non-malnourished residents, with the mean values of the two groups being around 202 mg/dL and 204.3 mg/dL, respectively (*p* = 0.305). 

### 4.6. Hemoglobin (Reference: >13 g/dL ♂/>12 g/dL ♀)

Female inhabitants had a MV of 12.12 SD: 1.65 g/dL, with values between 8.2 and 17.0 g/dL. Accordingly, 48.8% of the female residents had anemia of varying severity. Men had an MV of 12.1 SD: 1.55 g/dL, with values between 8.7 g/dL and 15.7 g/dL. Thus, 70.6% of the male subjects had anemia of varying severity. A clear but non-statistically significant correlation between lowered values and a higher mortality rate became apparent in anemic women (*p* = 0.081). In the male participants, on the other hand, there was no concordance at all (*p* = 0.97). Regardless of age and sex, malnourished patients had mean Hb values of 11.8 SD: 1.62 g/dL, which were significantly lower than those of non-malnourished study participants, with a mean value of 12.51 SD: 1.55 g/dL (*p* = 0.040). When comparing the Hb values of male inhabitants with and without DRM (11.9 g/dL SD: 1.55 and 12.2 SD: 1.57 g/dL, respectively) no significant correlation could be presented (*p* = 0.628) (see Table 10). Female malnourished patients, however, had statistically significant lower Hb values compared to non-malnourished patients (*p* = 0.029) (11.8 SD: 1.66 g/dL and 12.7 SD: 1.53 g/dL, respectively) (see Table 1.)

### 4.7. CRP (Reference: >3.0 (mg/dL) Were Considered Moderate to Severe Inflammation

In 29.2% of the 120 patients, the CRP value was above 3.0 mg/dL, and thus moderate to severe inflammation was present. In about 10%, i.e., 13 patients, the CRP value was between 0.0 and 0.1 mg/dL, while the maximum value was 14.4 mg/dL. The average CRP value was 2.54 mg/dL with an SD of 2.9 mg/dL (see Table 11). The average CRP level among residents with malnutrition was 1.95 mg/dL, compared to 2.07 mg/dL among non-malnourished residents. No significant association between malnutrition and elevated CRP was found in this study (*p* = 0.240).

## 5. Discussion

In this study, a very high mortality rate of 45.8% was observed within a period of 3 years and 7 months. In comparable studies, the median survival time was 2.2 years, while the 3-year mortality rate in comparable studies was around 50% [35].

An exceptionally high rate of malnourished nursing home residents was found, with a percentage of 37.9% within the first 3 months of admission and 41.1% after 6 months, respectively. However, according to the DGEM criteria, only 32.6% were considered to be malnourished after one year. 

When it comes to malnutrition, differences in the patient populations as well as the lack of uniform definitions and the absence of generally valid diagnostic criteria hinder detailed comparability to other international studies. However, most comparable studies indicate prevalence rates of around 15–45% [36,37,38], while other experts mention prevalence rates of malnutrition in nursing homes of up to 60% [39]. In this study, a definition of the DGEM guidelines was used, which is based on purely anthropometric data, such as BMI and weight history [40]. However, although weight and its progression are a very important indication of malnutrition that can be examined quickly, it is a much more complex subject.

Uniform diagnostic criteria for malnutrition, which additionally include laboratory data, are therefore indispensable.

Presumably, due to the small number of people included, no correlation with mortality could be shown, as was often the case in comparable studies [41,42]. More than half of nursing home residents were found to have low serum albumin saturation. Most comparable international studies performed in nursing homes describe prevalence rates of hypoalbuminemia between 19% and 55% [21,26,27,37]. Especially in recent work, the role of serum biomarkers, particularly serum albumin, in diagnosing or monitoring malnutrition is controversial, mainly due to its lack of specificity and long half-life (approximately 20 days) [43,44]. Thus, a connection between malnutrition and hypoalbuminemia was not found in this study.

Hypoalbuminemia may be the result of decreased production of albumin or increased loss of albumin via the kidneys, gastrointestinal tract, skin, or extravascular space. It may also be the result of increased catabolism of albumin, for instance, in inflammation, or a combination of two or more of these mechanisms [45]. In addition to the development of the disease itself, the condition of hypoalbuminemia could also be a consequence of the aging process, as albumin levels decrease with age [26].

However, the high prevalence of low albumin in elderly nursing home residents is particularly worrying, as this study also showed a clear correlation between hypoalbuminemia and increased mortality. Serum albumin remains the most cost-effective, common, and practical protein for assessing protein status and response to nutritional support and should therefore never be disregarded as part of a nutritional assessment [25].

Transferrin has also often been used as a marker of nutritional status in the recent past [46]. Serum levels decrease in the setting of severe malnutrition, but this marker is unreliable in the assessment of mild malnutrition in several recent studies [44]. According to the results of this work, transferrin does not serve as a suitable marker for nutritional status. However, transferrin is an important correlate of mortality in elderly patients [23], which can also be supported by the results of this study. According to the results of this study, hypoalbuminemia and low transferrin levels should be considered dangerous and problematic conditions on their own, especially in nursing home residents.

As described above, the etiology of malnutrition as well as the clinical presentation are diverse. Inflammation and emaciation are often observed simultaneously in the elderly, which is called malnutrition-inflammation-cachexia syndrome [47]. However, no significant association between malnutrition and elevated CRP values was found in this study. 

Equally noticeable were the high prevalence rates of anemic nursing home residents (48.8% of females, and 70.6% of male residents). Izaks et al. showed that anemic individuals aged 86 and over had a higher 5-year mortality rate than those with normal hemoglobin levels [48].

The most common incidence of anemia in older adults is related to malnutrition and is due to iron, folate, and/or vitamin B12 deficiency [49].

Of course, the laboratory parameters may be influenced by underlying chronic diseases; however, Zhiying Zhang et al. could show that BMI and several blood biochemicals, including albumin and hemoglobin, are useful biomarkers for adult malnutrition, even in the presence of chronic inflammation [50]. However, when analyzing malnutrition, laboratory parameters should always be considered in relation to anthropometric data and unplanned weight loss over time [51].

Finally, it is to be said that the difficulty of finding a generally valid definition is not only because malnutrition rarely is the primary reason for admission but also due to the high individual heterogeneity of malnutrition, nutrient deficiencies, or metabolic and nutritional disorders, especially in the elderly. 

Therefore, physicians play an important role, not only in the early detection and diagnosis of malnutrition, monitoring and possible therapy but also in the clinical application of personalized nutritional support for malnourished residents in the nursing home. Patients at particular risk of malnutrition with functional and/or cognitive impairment or dementia, swallowing problems, and depression can be identified and cared for separately by specially trained staff [41,52].

### Limitations

The findings of this study have to be seen in light of some limitations. 

Firstly, the sample size is relatively small, with 120 randomly selected patients who have been inpatients in the nursing home for at least one year. A larger sample size could definitely increase the power of similar studies.

A further major weakness of this study is the absence of laboratory data or inadequate collection or recording of anthropometric data. This is because patients are not routinely admitted by a doctor, as is usual in a hospital. 

In addition, the data on the cause of death could not be collected in this study, therefore, a more precise evaluation and contrast were not possible.

In most comparable studies today, the GLIM criteria are used to assess malnutrition in nursing homes [53]. In the present paper, these could not be carried out due to a lack of body composition measuring techniques. 

## 6. Conclusions

In this examination, more than one-third of nursing home residents were malnourished, whereas more than half of the residents were found to have low serum albumin or low transferrin saturation. Based on these worrying results, increased attention should be given to the urgent need for closer and personalized nutritional medical care and monitoring of nursing home residents, as well as targeted concomitant therapies for persons with difficulties in food intake, absorption, and/or metabolism. In addition to targeted nutrient supplementation, the choice of food and the environment also play a decisive role. Therefore, training programs for the entire team responsible would be highly beneficial. To fulfill these goals appropriately and comprehensively, it is essential to establish uniform and universally valid diagnostic criteria for the determination of malnutrition. In summary, we call for closer monitoring and recording of laboratory parameters and closer monitoring of weight by specially trained staff in nursing homes.

## Figures and Tables

**Table 1 nutrients-15-04367-t001:** Statistical comparison: “malnutrition = NO” vs. “malnutrition = YES”.

Parameters	Test	*p*-Wert
Age	MWU	0.384
Sex	Fisher	0.255
BMI	MWU	<0.001 **
BMI after 6 months	MWU	<0.001 **
BMI after 12 months	MWU	<0.001 **
CRP (mg/dL)	MWU	0.240
Serum albumin (g/L)	*t*-Test	0.788
Total protein (g/dL)	*t*-Test	0.120
Hemoglobin (g/dL)	*t*-Test	0.040 *
Transferrin (mg/dL)	MWU	0.305

** *p* < 0.01; * *p* < 0.05.

**Table 2 nutrients-15-04367-t002:** Patient status—last follow-up (08/2020).

Patient Status		n	%
	Inpatient	61	50.8%
	Discharged	4	3.3%
	Deceased	55	45.9%
	Total	120	100.00%

**Table 3 nutrients-15-04367-t003:** Patient status and malnutrition at the time of admission.

			Malnutrition		
			No	Yes	Total
Status	Inpatient	n	38	21	59
		%	64.41%	58.33%	62.11%
	Discharged	n	0	2	2
		%	0.00%	5.56%	2.11%
	Deceased	n	21	13	34
		%	35.59%	36.11%	35.79%
	Total	n	59	36	95
		%	100.00%	100.00%	100.00%

**Table 4 nutrients-15-04367-t004:** Malnutrition according to the DGEM Guidelines after 3 months.

Malnutrition after 3 Months		n	%	Deceased
	No	59	62.1%	0
	Yes	36	37.9%	
	Total	95	100.0%	

**Table 5 nutrients-15-04367-t005:** Malnutrition according to the DGEM Guidelines after 6 months.

Malnutrition after 6 Months		n	%	Deceased
	No	56	58.9%	0
	Yes	39	41.1%	
	Total	95	100.0%	

**Table 6 nutrients-15-04367-t006:** Malnutrition according to the DGEM Guidelines after 12 months.

Malnutrition after 12 Months		n	%	Deceased
	No	62	67.4%	3
	Yes	30	32.6%	
	Total	92	100.0%	

**Table 7 nutrients-15-04367-t007:** Serum albumin at time of admission.

Serum Albumin (g/L)		n	MV (g/L)	SD	Missing Data
	Inpatient/Discharged	64	35.0	4.0	1
	Deceased	55	31.7	4.5	0
Total		119	33.5	4.49	1

**Table 8 nutrients-15-04367-t008:** Malnutrition and serum Albumin at the time of admission.

Malnutrition		Serum Albumin (g/L)			
		n	MV	SD	Missing Data
	No	59	34.42	4.34	15
	Yes	36	34.2	3.66	
	Total	95	34.33	4.08	

**Table 9 nutrients-15-04367-t009:** Malnutrition and Transferrin (mg/dL).

Malnutrition		Transferrin (mg/dL)			
		n	MV	SD	Missing Data
	No	59	204.61	39.02	15
	Yes	36	202.22	48.23	
	Total	95	203.71	42.51	

**Table 10 nutrients-15-04367-t010:** Sex and Hb values at the time of admission.

Sex			n	MW	SD
female	Hb (g/dL)	Inpatient/Discharged	51	12.40	1.55
		Deceased	35	11.74	1.77
male	Hb (g/dL)	Inpatient/Discharged	14	12.08	1.74
		Deceased	20	12.10	1.46
Total			120	12.12	1.62

**Table 11 nutrients-15-04367-t011:** Malnutrition and CRP at the time of admission.

Malnutrition		CRP (mg/dL)			
		n	MV	SD	Missing Data
	No	59	2.07	2.56	15
	Yes	36	1.95	1.69	
	Total	95	2.02	2.26	

## Data Availability

Data is unavailable due to ethical restrictions.

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
