# Peer review of "The Urgent Need for Nutritional Medical Care in Geriatric Patients—Malnutrition in Nursing Homes"

_nutrients, 2023, doi:10.3390/nu15204367_

Round 1

Reviewer 1 Report

According to the authors from the database of 545 patients, 120 patients with a minimum length of stay of 12 months were randomly selected for this study.

Why were only 120 selected for the study? Were there any exclusion criteria except for the lengh of stay > 12 months? Detailed exclusion criteria should be provided, as well as information why specifically 120 patients were selected while data from a larger number of patients was available and the study could potentially be conducted on a larger sample.

The tables need refinement: titles should be more extensive and informative, explanations of symbols/abbreviations should be added. Authors alternately use a dot and a comma as a decimal separator - it should be unified.

The authors are transparent and clearly acknowledge the study limitations that statistical significance of the results cannot be assumed, which is essential for interpreting the results appropriately. However, increasing the sample size could enhance statistical power, making it more likely to detect significant effects. A larger sample size could help compensate for the limitations of a non-experimental design. Especially considering that the authors do not provide a reason for choosing such a small sample. Moreover, current indicators for assessing nutritional status are GLIM criteria.

The conclusions provided in the study do not provide an advancement of the current knowledge. They reflect well-established concerns related to malnutrition and nutritional care in elderly populations and nursing home settings.

Author Response

According to the authors from the database of 545 patients, 120 patients with a minimum length of stay of 12 months were randomly selected for this study. Why were only 120 selected for the study? Were there any exclusion criteria except for the lengh of stay > 12 months? Detailed exclusion criteria should be provided, as well as information why specifically 120 patients were selected while data from a larger number of patients was available and the study could potentially be conducted on a larger sample.

Answer: We appreciate your valuable comment that helped to enhance the clarity of the methods section. We have revised the methods section and once again clearly selected the recruited study patients based on the inclusion and exclusion criteria mentioned. Based on the criteria of the study protocol, 120 of 545 patients were ultimately declared eligible for inclusion, for whom the inclusion and exclusion criteria were precisely applied, moreover limitations part have been modified.

The tables need refinement: titles should be more extensive and informative, explanations of symbols/abbreviations should be added. Authors alternately use a dot and a comma as a decimal separator - it should be unified.

Answer: Thank you very much for this comment and we immediately apologize for the inconsistent punctuation used. We have tried to use the nomenclature and punctuation in a standardized manner throughout the entire manuscript; in addition, the tables, titles, names and figures have been revised and made clearer overall.

The authors are transparent and clearly acknowledge the study limitations that statistical significance of the results cannot be assumed, which is essential for interpreting the results appropriately. However, increasing the sample size could enhance statistical power, making it more likely to detect significant effects. A larger sample size could help compensate for the limitations of a non-experimental design. Especially considering that the authors do not provide a reason for choosing such a small sample. Moreover, current indicators for assessing nutritional status are GLIM criteria.

Answer: Thank you very much for this valuable comment. We completely agree with you and agree that an increase in the number of study patients included and thus a larger sample size would most likely improve the statistical power and increase the probability of achieving significant effects as a result of the evaluations. However, since 120 patients could be included in the study using strict inclusion and exclusion criteria, an increase in power was naturally not possible. We agree with you and would have liked to have included more patients in this study, but it must be said that a number of 120 patients is by no means a bad study population in such a setting. In the present paper, the GLIM-criteria were not carried out, due to a lack of body composition measuring techniques. The limitation part in the manuscript have been modified accordingly.

The conclusions provided in the study do not provide an advancement of the current knowledge. They reflect well-established concerns related to malnutrition and nutritional care in elderly populations and nursing home settings.

Answer: Thank you very much again for this comment, we totally agree with your consider. Conclusion and discussion part were modified accordingly.

Reviewer 2 Report

Careful attention to the limitations of this 'pilot' study with very small mortality numbers is needed.  

The significant figures for all the parameters need to be correct.  For example, the authors state a BMI of 22.88.  Given the variation in BMI estimates, the data should probably be presented as 22.8 or even 22.

Combining and shortening the tables would assist the reader in understanding the data.

On line 282, the authors indicate the weakness of the statistics.  Yet the statistics are emphasized throughout the paper.  The mortality data is particularly small and probably should be eliminated.  

Not bad.  Minor editing for clarity and conciseness would help. 

Author Response

Careful attention to the limitations of this 'pilot' study with very small mortality numbers is needed. The significant figures for all the parameters need to be correct. For example, the authors state a BMI of 22.88.  Given the variation in BMI estimates, the data should probably be presented as 22.8 or even 22. Combining and shortening the tables would assist the reader in understanding the data. On line 282, the authors indicate the weakness of the statistics.  Yet the statistics are emphasized throughout the paper.  The mortality data is particularly small and probably should be eliminated. 

Answer:  Thank you for taking time assessing this manuscript and for pointing this out. The figures for all parameters were corrected where necessary. The other concerns mentioned, such as the small sample size and small mortality data, were mentioned in the limitations.

Reviewer 3 Report

1. Malnutrition in older adults in nursing homes is well studied and a known issue in the United States but perhaps less studied in Germany or other countries.  The authors need to make a better case in the introduction that is specific to Germany and their nursing homes.  They also should address why all the lab values were selected based on literature that supports their use and linkage to outcomes, in particular, the CRP, transferrin, and hemoglobin.  

2. Lines 70--71: do not use the actual name of the nursing home since you also give dates of data collection and diagnoses you have too much personal identifiers.  Please change to something less specific.

3. From 75-91 it is very confusing exactly what you collected at each of the four times: baseline admission, 3, 6 and 12 months. You should present your variables for all time periods and not be lumping the results of each time period into one value which is how the tables read.

4. Lines 93-101 could be one paragraph and please state which variables were used in each of these tests. 

5. Lines 108-135 could also be consolidated into one section or paragraph labeled "measurements" or similar and include the information about the CRP, other variables in the results section.  Lines 116-119 is redundant to Lines 109-115.

6. You have a very high rate of missing data across all your variables, and fail to address this in your tables.  The very mortality of your sample is also of concern yet you do not speak to this as perhaps due to COVID?  Please list this as a weakness in your discussion area.

7. Results section: Why is the font different?  Also, please be sure the number of patients is included for each variable you mention.  None of the tables are helpful.  Please review some other similar studies on how to present your results.

8. LInes 219: The 37.9% malnourished rate is misleading if you are including what you found on admission, then at 3, 6 and 12 months.  Were these the same patients each time and no improvement?  Or only at the end of the study.  Please clarify.

9. All of your lab values can be effected by chronic diseases which diet alone cannot impact, that is, the disease must be well controlled by the medical care provider.  In addition, you completely omit the importance of the environment to promote good nutrition, that is adequately trained staff, sufficient staff, pleasing dining room, familiar foods, etc.  More labs and with use of supplements is not sufficient, it takes the whole team to improve the nutritional status of older adults in the nursing home.

English is fine.

Author Response

  1. Malnutrition in older adults in nursing homes is well studied and a known issue in the United States but perhaps less studied in Germany or other countries.  The authors need to make a better case in the introduction that is specific to Germany and their nursing homes.  They also should address why all the lab values were selected based on literature that supports their use and linkage to outcomes, in particular, the CRP, transferrin, and hemoglobin.  

Ad 1: Thank you very much for pointing this out. All concerns were considered, in addition, the introduction part was specially adapted to the requirements of the reviewers' comments mentioned above.

  1. Lines 70--71: do not use the actual name of the nursing home since you also give dates of data collection and diagnoses you have too much personal identifiers.  Please change to something less specific.

Ad 2: Thank you. The name of the nursing home has been removed.

  1. From 75-91 it is very confusing exactly what you collected at each of the four times: baseline admission, 3, 6 and 12 months. You should present your variables for all time periods and not be lumping the results of each time period into one value which is how the tables read.

Ad 3: We agree that the presented paragraph was confusing. The only data we were able to collect from our patients at regular intervals (after 3, 6 and 12 months) were height and weight. Blood samples were only collected at the time of admission.  The tables were edited accordingly, and the methodology was adapted.

  1. Lines 93-101 could be one paragraph and please state which variables were used in each of these tests.

Ad 4. : We appreciate your valuable comment that helped to enhance the clarity of the methods section, which was specially adapted to the requirements of the reviewers' comments mentioned above.

  1. Lines 108-135 could also be consolidated into one section or paragraph labeled "measurements" or similar and include the information about the CRP, other variables in the results section.  Lines 116-119 is redundant to Lines 109-115.

Ad 5: Thank you very much for this comment. According to the reviewer's suggestion, a new paragraph with the subtitle "Measurements" was implemented, which contains relevant information. The descriptions of the other variables have been moved to the results part.

  1. You have a very high rate of missing data across all your variables and fail to address this in your tables.  The very mortality of your sample is also of concern, yet you do not speak to this as perhaps due to COVID?  Please list this as a weakness in your discussion area.

Ad 6: Thank you for your suggestion and your very important comment. The missing data was addressed in the tables and according to the suggestions of the reviewers the mortality rates were added into the discussion part.

  1. Results section: Why is the font different?  Also, please be sure the number of patients is included for each variable you mention.  None of the tables are helpful. Please review some other similar studies on how to present your results.

Ad 7: Thank you very much for this valuable comment. In the course of writing the manuscript, we may not have noticed that another font was used in this section of the manuscript. We're sorry, and of course the font has been adjusted to fit the journal and the rest of the manuscript. In addition, according to the reviewers' specifications, the number of patients was stated for each variable mentioned. The tables were subsequently updated according to the suggestions so that the relevant data is better reflected.

  1. LInes 219: The 37.9% malnourished rate is misleading if you are including what you found on admission, then at 3, 6 and 12 months.  Were these the same patients each time and no improvement?  Or only at the end of the study.  Please clarify.

Ad 8: Thank you very much for your comment. We have revised this part and have tried to write this paragraph in more detail, so that the message and content regarding the topic appears clearer. Corresponding changes were made in the discussion section of the manuscript.

  1. All of your lab values can be effected by chronic diseases which diet alone cannot impact, that is, the disease must be well controlled by the medical care provider.  In addition, you completely omit the importance of the environment to promote good nutrition, that is adequately trained staff, sufficient staff, pleasing dining room, familiar foods, etc.  More labs and with use of supplements is not sufficient, it takes the whole team to improve the nutritional status of older adults in the nursing home.

Ad 9: Thank you very much for this comment and the valuable analysis of the situation. We share these concerns and have of course addressed this and tried to make our statements clearer in this regard. Moreover, we have adapted the discussion and conclusion part accordingly.

Round 2

Reviewer 1 Report

Thank you for providing revised version of the manuscript. After the improvements the quality of the article has increased.

Reviewer 2 Report

improved

Reviewer 3 Report

All concerns were addressed and this manuscript is much improved.